# TextField3D: Towards Enhancing Open-Vocabulary 3D Generation with Noisy Text Fields

**Tianyu Huang**[1,3] **Yihan Zeng**[2] **Bowen Dong**[1] **Hang Xu**[2] **Songcen Xu**[2]
**Rynson W. H. Lau**[3] **Wangmeng Zuo**[1†]
[1]Harbin Institute of Technology  [2]Huawei Noah's Ark Lab  [3]City University of Hong Kong
`tyhuang@stu.hit.edu.cn,{zengyihan2,xu.hang,xusongcen}huawei.com,`
`cndongsky@gmail.com,rynson.lau@cityu.edu.hk,wmzuo@hit.edu.cn`

## Abstract

Generative models have shown remarkable progress in 3D aspect. Recent works learn 3D representation explicitly under text-3D guidance. However, limited text-3D data restricts the vocabulary scale and text control of generations. Generators may easily fall into a stereotype concept for certain text prompts, thus losing open-vocabulary generation ability. To tackle this issue, we introduce a conditional 3D generative model, namely TextField3D. Specifically, rather than using the text prompts as input directly, we suggest to inject dynamic noise into the latent space of given text prompts, *i.e.*, Noisy Text Fields (NTFs). In this way, limited 3D data can be mapped to the appropriate range of textual latent space that is expanded by NTFs. To this end, an NTFGen module is proposed to model general text latent code in noisy fields. Meanwhile, an NTFBind module is proposed to align view-invariant image latent code to noisy fields, further supporting image-conditional 3D generation. To guide the conditional generation in both geometry and texture, multi-modal discrimination is constructed with a text-3D discriminator and a text-2.5D discriminator. Compared to previous methods, TextField3D includes three merits: 1) large vocabulary, 2) text consistency, and 3) low latency. Extensive experiments demonstrate that our method achieves a potential open-vocabulary 3D generation capability.

## 1 Introduction

3D creative contents are in significantly increased demand for a wide range of applications, including video games, virtual reality, and robotic simulation. However, manually creating 3D content is a time-consuming process that requires a high level of expertise. To achieve automatic generation, previous works (Niemeyer & Geiger, 2021; Gu et al., 2021; Chan et al., 2022; Gao et al., 2022) attempt on several 3D representations like Voxel, NeRF, and SDF. Nonetheless, objects from these methods are unconditionally generated in a single category, which can hardly be applied in practice.

With the success of text-to-image generative models (Ramesh et al., 2021; 2022; Rombach et al., 2022), text prompts become a flexible variable to achieve open-vocabulary generative capability. Similar to 2D vision, text-to-3D generation has recently aroused great interest. Current research can be generally separated into two strategies. One (Jain et al., 2022; Mohammad Khalid et al., 2022; Poole et al., 2022; Lin et al., 2022) optimizes differentiable 3D representations with vision-language (V-L) pre-trained knowledge, *e.g.*, DreamFields (Jain et al., 2022) is optimized from knowledge of the V-L pre-trained model CLIP (Radford et al., 2021), and DreamFusion (Poole et al., 2022) is guided by a diffusion model Imagen (Saharia et al., 2022). Since no 3D data are involved in training, these methods can easily fall into artifacts, facing serious 3D consistency problems. Meanwhile, the cost of optimizing 3D representations is extremely high, in terms of both time and computational resources. The first strategy seems not to be the ultimate solution for 3D generation. The other (Nichol et al., 2022; Cheng et al., 2023; Wei et al., 2023; Sanghi et al., 2023; Jun & Nichol, 2023) directly supervises 3D generators with paired text-3D data. These methods allow real-time generation but are extremely restricted by the scale of available 3D data. Nichol et al. (2022) and Jun & Nichol

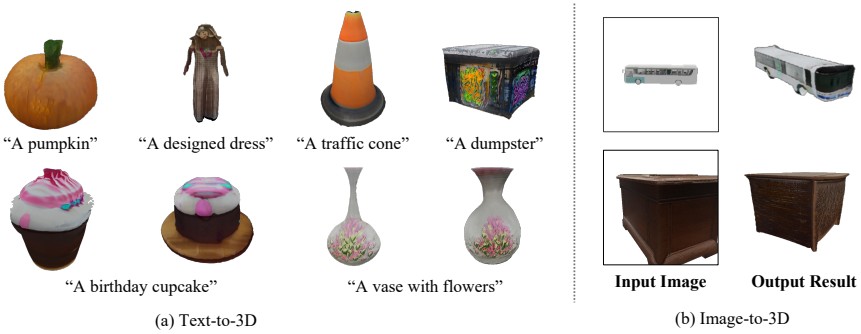

(a) Text-to-3D                                              (b) Image-to-3D

Figure 1: TextField3D is capable of text-to-3D and image-to-3D tasks. In particular, all the text prompts are from (Nichol et al., 2022; Jun & Nichol, 2023), our method can generate diverse and reasonable results, exhibiting a potential open-vocabulary capability.

(2023) collect millions of text-3D pairs, which is still thousands of times smaller than the scale of V-L data. As such, the diversity of generated output and the complexity of text input are incompatible with V-L optimized methods. Therefore, here is the question: With limited data, can we train a real-time 3D generator with the potential towards open-vocabulary content creation?

The common practice to address data scarcity is introducing V-L pre-trained knowledge to generators. Specifically, 3D generative latent codes are initialized with pre-trained models. Cheng et al. (2023) adopt BERT (Devlin et al., 2018)/CLIP (Radford et al., 2021) encoders for text/image-condition tasks. Sanghi et al. (2023) propose to train a transformer module with CLIP image embedding and replace it with text embedding at inference. Wei et al. (2023) further attempt to align text prompts with rendered images to improve generative text control. However, one key problem remains challenging: how to map limited 3D content to comprehensive V-L pre-trained concepts. Some recent 3D representation learning works Huang et al. (2023); Liu et al. (2023a) align generative latent space based on limited categories and text prompts, resulting in a close set understanding.

To tackle this issue, we intend to expand the expression range of 3D latent space. We observe that 3D latent code can easily fall into stereotype concepts when trained in a small data scale with clean data and templated prompts. For example, a simple prompt "**a chair**" can represent any type of chair, tall or short, with or without armrests. However, if supervised on a specific type of chair, the generator may tend to produce this fixed shape only, losing diversity. Based on this observation, we introduce a conditional 3D generative model, dubbed TextField3D. TextField3D maps limited 3D data to dynamic fields of V-L concepts, which are named as Noisy Text Fields (NTFs). Specifically, we assume that there is a dynamic range in the latent mapping of text-to-3D, which can be formulated as an injected noise to the text embeddings. We thus propose an NTFGen module, where text latent code is formulated based on the field of noisy text embeddings. To support image conditional generation, we further propose an NTFBind module for binding image features to NTFs and keeping the consistency of multi-view conditions. A view-invariant image latent code is derived for the image-to-3D task. Furthermore, multi-modal discrimination is constructed for the supervision of 3D generation, in which we propose a text-3D discriminator to guide geometry generation and a text-2.5D discriminator to refine texture details.

Compared to previous methods, TextField3D simultaneously possesses three advantages: large vocabulary, text consistency, and low latency. Extensive experiments are conducted on various categories and complicated text prompts, which exhibit the open-vocabulary potential of our model. We further employ several metrics to evaluate the generation quality and text consistency, demonstrating the effectiveness of our newly proposed modules.

Our contributions can be summarised as follows:

- We introduce a conditional 3D generative model TextField3D, in which limited 3D data is mapped to textual fields with dynamic noise, namely Noisy Text Fields (NTFs).

- We propose NTFGen and NTFBind to manipulate general latent codes for conditional generation, and a multi-modal discriminator to guide the generation of geometry and texture.

- Extensive experiments present an open-vocabulary potential of our proposed method, in terms of large vocabulary, text consistency, and low latency.

## 2 RELATED WORK

Driven by the increasing demand for 3D creative contents, text-to-3D generation has witnessed rapid improvement recently. We can generally split current studies into two categories: V-L optimized methods and 3D supervised methods.

**V-L Optimized Methods.** With the success of V-L pre-trained models, various works (Jain et al., 2022; Mohammad Khalid et al., 2022; Poole et al., 2022; Lin et al., 2022; Chen et al., 2023; Wang et al., 2023) leverage V-L pre-trained knowledge to optimize implicit functions of 3D representations. Jain et al. (2022) supervise the generation of a NeRF representation by the CLIP guidance of given text prompts. Poole et al. (2022) introduce Score Distillation Sampling (SDS) to a NeRF variant, adopting diffusion models to optimize 3D renderings. Based on SDS, Wang et al. (2023) further propose Variational Score Distillation (VSD), treating the 3D content of a given textual prompt as a random variable. Albeit approaching open-vocabulary generation, these methods highly depend on expensive and time-consuming optimization procedures and may easily produce 3D artifacts like multi-face and paper-thin outputs, which is not a practical solution for 3D generation.

**3D supervised Methods.** 3D supervised methods train generators with 3D data. Contrary to V-L optimized methods, they present an efficient way to generate solid 3D content. These works (Achlioptas et al., 2018; Kosiorek et al., 2021; Chen & Wang, 2022) are basically in an encoder-decoder architecture. For example, Kosiorek et al. (2021) proposes a variational autoencoder, encoding rendered views of scene representations as a generative condition. However, generations are restricted by the scarcity of 3D data. To allow various generative conditions, some recent works (Sanghi et al., 2022; Mittal et al., 2022; Fu et al., 2022; Cheng et al., 2023; Wei et al., 2023; Sanghi et al., 2023) attempt to combine pre-trained knowledge to the latent encoding, while the improvement is limited. Nichol et al. (2022) and Jun & Nichol (2023) collect several millions 3D data for training large-vocabulary generators, which is still far from the data scale of 2D diffusion.

In this work, we aim to enhance the mapping of limited 3D data and comprehensive V-L pre-trained knowledge, presenting an efficient way to approach open-vocabulary 3D generation.

## 3 PRELIMINARIES

### 3.1 3D GENERATIVE MODEL

Regardless of those V-L optimized methods, a 3D generative model takes latent codes to the generator and supervises corresponding generated contents with 3D data. GET3D (Gao et al., 2022) proposes a 3D generative adversarial network (GAN) that embeds a random vector to the latent space of geometry and texture. MeshDiffusion (Liu et al., 2023b) learns to denoise a simple initial distribution into the target 3D objects with a diffusion model. SDFusion (Cheng et al., 2023) leverages a vector quantised-variational autoencoder (VQ-VAE) to compress 3D shapes into latent space for further generations. Not losing generality, we can split 3D generation training into three components, *i.e.*, a 3D latent code $\mathbf{w}$, a generator $G$ in differentiable 3D representations, and a 3D-related supervision $\nabla_{3D}G(\mathbf{w})$.

### 3.2 CONDITIONAL 3D GENERATION

To allow condition inputs for the generation, the common strategy is embedding the generative latent code with corresponding inputs. In 2D text-to-image synthesis (Ramesh et al., 2021; 2022; Rombach et al., 2022), generators are trained to minimize the objective $\nabla_{2D}G(\mathbf{w})$ with large-scale V-L data, achieving an open-vocabulary performance. Unfortunately, reaching such a tremendous scale for 3D datasets is challenging. Saharia et al. (2022) suggest that pre-trained text or V-L models may encode meaningful representations relevant for the generation task. They initialize latent codes with freezed BERT (Devlin et al., 2018), T5 (Raffel et al., 2020), and CLIP (Radford et al., 2021), which may alleviate data scarcity to some extent. Therefore, some text-to-3D methods (Sanghi et al., 2022; Mittal et al., 2022; Fu et al., 2022; Cheng et al., 2023; Wei et al., 2023; Sanghi et al., 2023) utilize pre-trained knowledge to embed latent vectors $\mathbf{w}$. Take a textual input $t$ as an example: The conditional latent code $\mathbf{w}_t$ can be revised as $\mathbf{w}_t = f(z, \mathbf{t})$, where $\mathbf{t}$ is pre-trained text features encoded from $t$, and $f$ is a mapping network to embed text features $\mathbf{t}$ with noise distributions $z$.

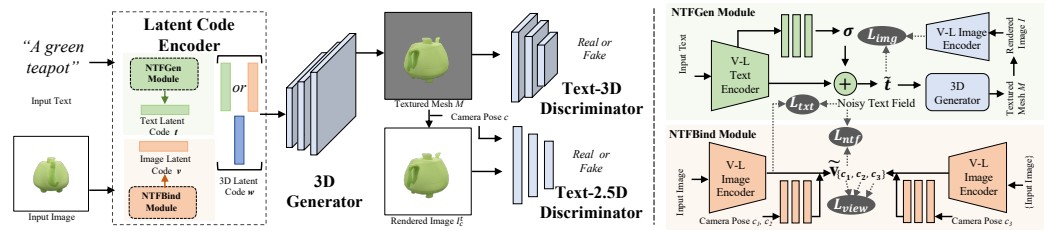

Figure 2: The overall framework of TextField3D. The NTFGen module and the NTFBind module are proposed to encode conditional latent codes in noisy text fields, which are fed to a 3D generator. Multi-modal discrimination is then exploited to supervise the generation process, including a text-3D discriminator and a text-2.5D discriminator.

# 4  METHODOLOGY

Albeit introducing open-vocabulary knowledge from V-L pre-trained models as guidance, previous 3D generative models (Cheng et al., 2023; Wei et al., 2023; Sanghi et al., 2023) still struggle to exhibit promising open-vocabulary capability. In this work, we present TextField3D as a feasible solution to approach open-vocabulary generation. To this end, we introduce Noisy Text Fields (NTFs) to boost the latent mapping between V-L concepts and 3D representations. Meanwhile, we propose multi-modal discrimination to enhance the supervision of 3D generation. The overall framework is illustrated in Figure 2. We provide detailed explanations in the following.

## 4.1  LEARNING GENERAL LATENT SPACE FOR 3D REPRESENTATION

Current available 3D data is limited to a small range of categories, which is unable to cover comprehensive concepts of V-L pre-trained knowledge. As shown in Figure 3, previous 3D latent code is trained with clean data and templated prompts, falling into a stereotype concept of certain prompts in training data. Therefore, it is a challenging task to align 3D latent space with the open-vocabulary embedding space of V-L pre-trained models.

In order to learn a general latent code from limited 3D data, we have to map 3D data to an equivalent expression range of pre-trained concepts. There are some previous studies (Nukrai et al., 2022; Gu et al., 2022) addressing the issue of imbalanced data between multiple modalities. They attempt to solve V-L tasks with text-only training. An injected noise $n \sim N(0, \epsilon)$ is attached to the embedded text, so that image features can be included in the textual space. In our case, text prompts and 3D objects have a one-to-many correspondence, *e.g.*, "a chair" can represent all kinds of chairs in Figure 3. Accordingly, we can introduce a similar noise to the corresponding 3D latent space.

**Noisy Text Latent Code.**  Considering the scarcity of available text-3D data, we attach a textual range to 3D latent space following (Nukrai et al., 2022; Gu et al., 2022). However, we observe that the one-to-many situation is more obvious in 3D. Different from 2D realistic images, rendered images of 3D data present a single 3D object on a clean background. The diversity can be easily constrained if a strict text prompt is given like "a tall wooden chair without armrests". Therefore, we propose a dynamic noise injection to adjust the range of text embeddings $\mathbf{t}$:

$$\widetilde{\mathbf{t}} = \mathbf{t} + N(0, \boldsymbol{\sigma}), \; where \; \boldsymbol{\sigma} = f_{txt}(\mathbf{t}), \boldsymbol{\sigma} \in (\epsilon_1, \epsilon_2), \tag{1}$$

where $f_{txt}$ is a non-linear network. $\epsilon_1$ and $\epsilon_2$ are hyperparameters, which are set as 0.0002 and 0.016. $\widetilde{\mathbf{t}}$ is a textual field injected with dynamic noise, which is regarded as the **Noisy Text Field** (NTF) of $\mathbf{t}$. Latent code can then be calculated as $\mathbf{w} = f(z, \widetilde{\mathbf{t}})$ in NTFs, while we observe that the mapping network $f$ weakens the generative control of $\widetilde{\mathbf{t}}$. In text-to-image synthesis, Härkönen et al. (2022) and Sauer et al. (2023) also notice the tendency that the input random vector $z$ can dominate over text embeddings $\mathbf{t}$. Following them, we propose a late concatenation, bypassing the mapping networks for text embeddings $\widetilde{\mathbf{t}}$. We thus have the noisy text latent code $\widetilde{\mathbf{w}}_t = [\mathbf{w}, \widetilde{\mathbf{t}}]$.

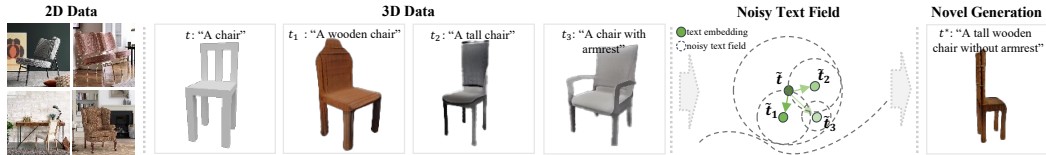

Figure 3: The motivation of Noisy Text Field. Previous works may easily form stereotype concepts of training data but have no idea about out-of-distribution prompts. Differently, the Noisy Text Field can model the semantic correspondences among cases.

To optimize the alignment of latent code and correspondingly generated contents, the common practice is minimizing the cosine distance between text features and image features,

$$\mathcal{L}_{nce}(I) = -\log \frac{\exp(\widetilde{\mathbf{t}} \cdot [f_{vis}(I_c^t)]^T/\tau)}{\exp(\widetilde{\mathbf{t}} \cdot [f_{vis}(I_c^t)]^T/\tau) + \sum_{\bar{t} \neq t} \exp(\widetilde{\mathbf{t}} \cdot [f_{vis}(I_c^{\bar{t}})]^T/\tau)}, \qquad (2)$$

where $f_{vis}$ is a pre-trained visual encoder. The image $I_c^t$ is rendered by a renderer $R$ at the viewpoint $c$, which is defined as $I_c^t = R(G(\widetilde{\mathbf{w}}_t), c)$. However, $\mathcal{L}_{nce}(I)$ is a generation-based objective to match the generated content with the ground-truth text prompt. We further propose a condition-based objective $\mathcal{L}_{nce}(\hat{I})$, enhancing the alignment of latent code in NTFs with ground-truth rendered images $\hat{I}$, The final objective for generating noisy textual fields is $\mathcal{L}_{gen} = \mathcal{L}_{nce}(I) + \mathcal{L}_{nce}(\hat{I})$. The module is thus named as NTFGen.

**View-Invariant Image Latent Code.** To allow image conditional 3D generation, previous works (Cheng et al., 2023; Gupta et al., 2023) replace text embeddings $\mathbf{t}$ with visual embeddings $\mathbf{v}$ as generative conditions. However, we argue that image conditions are not equivalent to text conditions in the 3D domain, as images provide additional viewpoint information. In order that generated objects can keep consistent with image conditions at different views, we learn a view-invariant image latent code $\widetilde{\mathbf{v}}_c^t = f_{view}(f_{vis}(\hat{I}_c^t), c)$, where $f_{view}$ is a non-linear network conditioned with camera poses. Given an input latent code $\widetilde{\mathbf{v}}_{c_1}^{t_1}$, we randomly select its positive latent code $\widetilde{\mathbf{v}}_{c_2}^{t_1}$ and negative latent code $\widetilde{\mathbf{v}}_{c_3}^{t_2}$. The view-invariant objective $\mathcal{L}_{view}$ expects to bind feature spaces with the same text prompt, which is formulated as $\mathcal{L}_{view} = \left\| \widetilde{\mathbf{v}}_{c_1}^{t_1} - \widetilde{\mathbf{v}}_{c_2}^{t_1} \right\|_2 - \left\| \widetilde{\mathbf{v}}_{c_1}^{t_1} - \widetilde{\mathbf{v}}_{c_3}^{t_2} \right\|_2$. Meanwhile, we bind the corresponding generations $I_c^t$ to noisy text fields $\widetilde{\mathbf{t}}$ by an NTF binding loss $\mathcal{L}_{ntf}(I_c^t)$,

$$\mathcal{L}_{ntf}(I_c^t) = -\log \frac{\exp(\widetilde{\mathbf{t}} \cdot [f_{view}(f_{vis}(I_c^t), c)]^T/\tau)}{\exp(\widetilde{\mathbf{t}} \cdot [f_{view}(f_{vis}(I_c^t), c)]^T/\tau) + \sum_{\bar{t} \neq t} \exp(\widetilde{\mathbf{t}} \cdot [f_{view}(f_{vis}(I_c^{\bar{t}}), c)]^T/\tau)} \qquad (3)$$

The final binding objective is $\mathcal{L}_{bind} = \mathcal{L}_{view} + \mathcal{L}_{ntf}$, and we regard the module as NTFBind.

## 4.2 TEXTURED MESH GENERATION WITH MULTI-MODAL DISCRIMINATION

After designing general latent code for 3D generation with injected noise, we then discuss the 3D generative model we used in TextField3D. Previous textured mesh generative models (Chan et al., 2022; Gao et al., 2022; Gupta et al., 2023) have shown remarkable progress in terms of generation quality. In particular, GET3D (Gao et al., 2022) attaches a texture field (Oechsle et al., 2019) to a differentiable 3D representation DMTet (Shen et al., 2021), efficiently producing high-quality textured meshes. Following it, we feed the noisy texture fields to formulate a conditional generator toward open-vocabulary 3D content. Furthermore, to better guide the generation process, we introduce multi-modal discrimination and build a generative adversarial training framework. In the following, we illustrate each component in this framework.

**Textured Mesh Generator.** Given two random vectors $z_1$ and $z_2$, GET3D adopts two non-linear mapping networks $f_{geo}$ and $f_{tex}$ for intermediate latent vectors $\mathbf{w}_1 = f_{geo}(z_1)$ and $\mathbf{w}_2 = f_{tex}(z_2)$, which are then used for the generation of 3D shapes and texture. By embedding noisy textual fields $\widetilde{\mathbf{t}}$ and bypassing mapping networks $f_{geo}$ and $f_{tex}$, the generation process is formulated as $G([\mathbf{w}_1, \widetilde{\mathbf{t}}], [\mathbf{w}_2, \widetilde{\mathbf{t}}])$.

**Text-2.5D Discriminator.** The combination of rendered images and corresponding camera poses is widely used in GAN-based 3D supervision (Gu et al., 2021; Chan et al., 2022; Gao et al., 2022).

Rendered images can provide detailed visual information that is beneficial for texture generation. Since 3D geometry is implicitly presented by camera poses, we regard such discrimination as 2.5D supervision. To further supervise the text consistency, we concatenate the camera pose $c$ with $\widetilde{\mathbf{t}}$ as a new discriminative condition. The text-2.5D discriminative objective is formulated as,

$$\mathcal{L}(D_x, G) = \mathbb{E}_{t \in T} \, g(D_x(I_c^t, [c, \widetilde{\mathbf{t}}])) + \mathbb{E}_{t \in T, \hat{I}_c^t \in \hat{I}} \, (g(-D_x(\hat{I}_c^t, [c, \widetilde{\mathbf{t}}])) + \lambda \left\| \nabla D_x(\hat{I}_c^t) \right\|_2^2), \quad (4)$$

where $g(x)$ is defined as $g(x) = -log(1 + \exp(x))$. $\hat{I}$ is the ground-truth rendered image set. $\lambda$ is a hyperparameter. $x$ is related to either RGB image or silhouette mask. Following GET3D, we apply the supervision of both RGB and mask.

**Text-3D Discriminator.** Although 2.5D discrimination implies 3D spatial information, it is difficult to represent geometric details of objects such as normal, smoothness, and so on. Some recent works (Liu et al., 2023b; Cheng et al., 2023; Gupta et al., 2023) have attempted to learn the shape prior directly from 3D data. Similarly, we exploit sampled point clouds to discriminate the generative quality. Given a generated mesh object, we uniformly sample point clouds $p$ over its surface. A discriminator architecture similar to 2.5D is then adopted, while we adjust its mapping network to PointNet (Qi et al., 2017). We have the text-3D discriminative objective $\mathcal{L}(D_{pc}, G)$,

$$\mathcal{L}(D_{pc}, G) = \mathbb{E}_{t \in T} \, g(D_{pc}(p^t, \mathbf{t})) + \mathbb{E}_{t \in T, \hat{p}^t \in \hat{P}} \, (g(-D_{pc}(\hat{p}^t, \mathbf{t})) + \lambda \left\| \nabla D_{pc}(\hat{I}_c^t) \right\|_2^2), \quad (5)$$

where $p^t$ and $\hat{p}^t$ are the generated and the ground-truth point clouds with text prompt $t$, respectively. $\hat{P}$ is the ground-truth point cloud set.

### 4.3 OVERALL TRAINING OBJECTIVE

Finally, we illustrate the overall training objective $\mathcal{L}$ of TextField3D, which is defined as follows:

$$\mathcal{L} = \mathcal{L}(D_{img}, G) + \mathcal{L}(D_{mask}, G) + \lambda_{pc}\mathcal{L}(D_{pc}, G) + \lambda_{gen}\mathcal{L}_{gen}, \quad (6)$$

where the loss weight parameters $\lambda_{pc} = 0.01$ and $\lambda_{gen} = 2$. The NTFBind module is additionally trained by the binding objective $\mathcal{L}_{bind}$. During its training, we freeze noisy texture fields $\widetilde{\mathbf{t}}$.

## 5 EXPERIMENTS

### 5.1 DATASET

We train TextField3D with a large-scale 3D dataset Objaverse (Deitke et al., 2022), which collects over 800k 3D objects from various categories. We select around 175k objects among them, which are qualified for the training. To prepare training data, we render RGBA images of 3D objects from random camera viewpoints with blender engines (Community, 2018) and annotate textual descriptions to them with the latest image captioning models (Li et al., 2023). For ablation studies, a clean 3D dataset ShapeNet (Chang et al., 2015) is used to evaluate the generative quality. ShapeNet contains over 50k clean 3D objects from 55 categories. We can accurately divide the training and test sets based on categories, which is the reason why we choose ShapeNet for the ablation studies. We filter out objects without texture, leaving 32,863 3D textured shapes. The filtered data are split into training, validation, and testing sets in a ratio of 7:1:2. We additionally annotate ShapeNet with MiniGPT-4 (Zhu et al., 2023). Please refer to Appendix B for details of data preparation.

### 5.2 IMPLEMENTATION DETAILS

For Objaverse, we train TextField3D with the full set and evaluate it with open-vocabulary text prompts. For ShapeNet, we select the best-performing models on the validation set for the evaluation. The total training time is around 3 days and 1 day with 8 V100 GPUs, respectively. Following GET3D, we use a StyleGAN (Karras et al., 2019) discriminator architecture. We use Adam (Kingma & Ba, 2014) optimizer and initialize the learning rate to $0.002$. The training batch size is 64. We use CLIP's ViT-B/32 to embed text/image inputs and freeze them on training. We sample 8,192 points for each mesh object. And the image resolution is $512 \times 512$ for both rendering and generation.

## 5.3 EXPERIMENTAL RESULTS

We train TextField3D on the Objaverse dataset, which is captioned with BLIP-2 (Li et al., 2023). Four state-of-the-art methods are then selected for comparison: DreamFields (Jain et al., 2022), DreamFusion (Poole et al., 2022), Point-E (Nichol et al., 2022), and Shap-E (Jun & Nichol, 2023). Quantitative and qualitative results are presented in the following.

**Quantitative Results.** To verify open-vocabulary generation ability, Jain et al. (2022) propose an evaluation set, collecting 153 captions from the Common Objects in Context (COCO) dataset (Lin et al., 2014). We follow the experiment setting: for each prompt, we render an image from the corresponding generated object, with a fixed camera pose at a $45°$ angle of elevation and a $30°$ angle of rotation. Table 1 reports the retrieval precision with CLIP (CLIP R-Precision). Our 1-shot results are better than Point-E and Shap-E, even though TextField3D is trained in a much smaller data scale. Moreover, since TextField3D generates 3D objects much faster than other methods, we try eight more times for each prompt and select the most similar image out of 9 (according to the cosine similarity with CLIP text features). The results (TextField3D, 9-shot) can reach a higher level, almost close to SDS methods. These results demonstrate that TextField3D is more generalized than previous supervised methods (Point-E and Shap-E) with much fewer training data and achieves a comparable open-vocabulary capability with V-L optimized methods by a much faster inference.

**Qualitative Results** We select seven representative text prompts for the text-to-3D generation task, including three single nouns, two compound nouns, and two adjectival nouns. As shown in Figure 4, the generation quality of DreamFields and Point-E is relatively low. DreamFusion has several paper-thin cases, *i.e.*, rat, chess piece, and brick chimney, as it is optimized V-L pre-trained knowledge rather than 3D shape prior. The view consistency is another problem for DreamFusion, *e.g.*, the generated teapot has multiple spouts. Shap-E is a 3D-optimized method that is similar to our TextField3D. However, it fails to generate a common rat, which raises questions about its open-vocabulary capability. Moreover, Shap-E performs poorly in text prompts with simple attribute binding. The chimney does not have a brick texture, and the teapot is not uniformly filled with green color. The results of our model TextField3D are basically consistent with text descriptions.

We also conduct image-to-3D generation, comparing with Point-E and Shap-E in Figure 5. "TextField3D" denotes replacing text embeddings $\widetilde{\mathbf{t}}$ with image features. "+NTFBind" denotes using view-invariant image latent code $\widetilde{\mathbf{v}}_c^t$. In (a), we provide part of a clay pot in a bottom view as input. Shap-E can generate a similar shape but the color is lighter than the ground-truth image. TextField3D can reproduce the color, while the generated pot neck is too long due to the interference of the input view. The shape and texture are generated precisely only when our NTFBind module is employed. The front view of a donkey in (b) is much harder to reconstruct, as it looks quite different in multiple views. Shap-E fails to reproduce the donkey, generating extra legs in a lighter color. The donkey generated by TextField3D is better than Shap-E's, but it still has obvious artifacts at the neck and back. The reproduction with our NTFBind has the best visual result.

## 5.4 ABLATION STUDIES

**NTFGen Module and Multi-Modal Discriminator.** To demonstrate the effectiveness of these two proposed modules, we compare four conditional generation solutions, *i.e.*, (a) embedding pre-trained V-L features into GET3D, which is similar to SDFusion (Cheng et al., 2023); (b) further

Table 1: CLIP R-Precision on COCO evaluation prompts.

| Method | ViT-B/32 | ViT-L/14 | Latency | Data Scale |
|---|---|---|---|---|
| DreamFields | 78.6 | 82.9 | $\sim$ 200 V100-hr | - |
| DreamFusion | 75.1 | 79.1 | $\sim$ 12 V100-hr | - |
| Point-E (300M, text-only) | 33.6 | 35.5 | 24 V100-sec | > 1M |
| Shap-E (300M, text-only) | 37.8 | 40.9 | 13 V100-sec | > 1M |
| Point-E (300M) | 40.3 | 45.6 | 1.2 V100-min | > 1M |
| Point-E (1B) | 41.1 | 46.8 | 1.5 V100-min | > 1M |
| Shap-E (300M) | 41.1 | 46.4 | 1.0 V100-min | > 1M |
| TextField3D (ours, 1-shot) | 45.8 | 49.0 | $\sim$ 6.9 V100-sec | $\sim$ 175K |
| TextField3D (ours, 9-shot) | 63.4 | 67.3 | $\sim$ 1.0 V100-min | $\sim$ 175K |

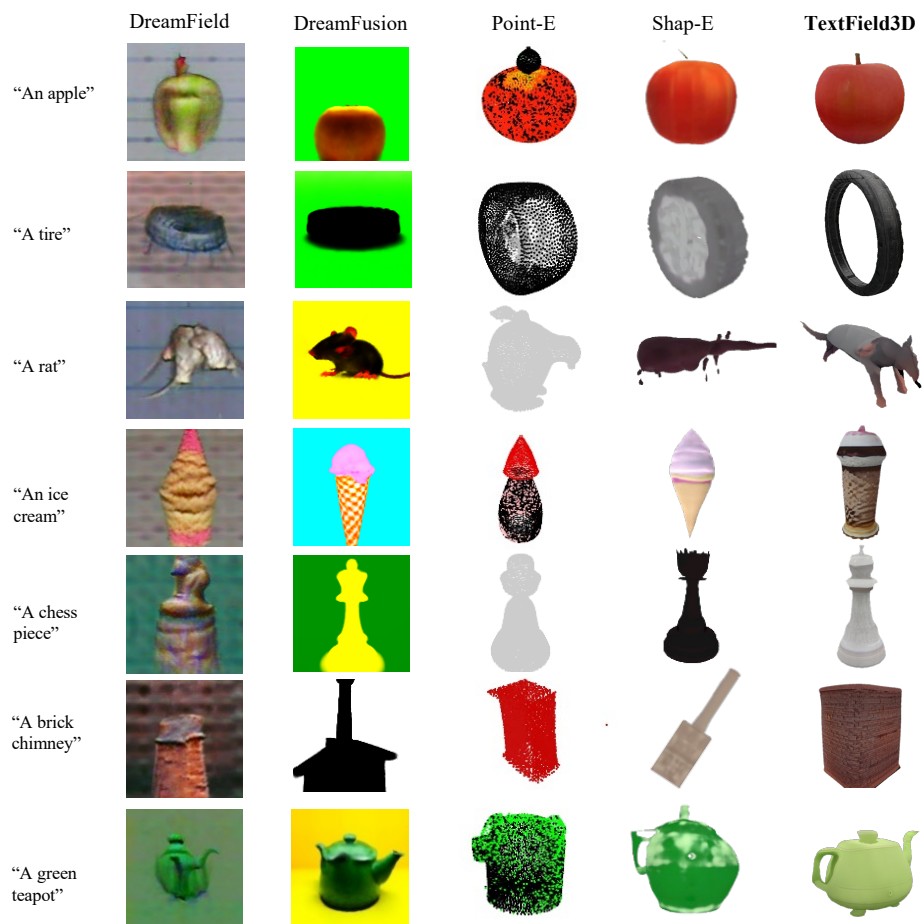

Figure 4: Text-to-3D generation. We use seven representative text prompts to exhibit open-vocabulary generative capability, including single nouns, compound nouns, and adjectival nouns.

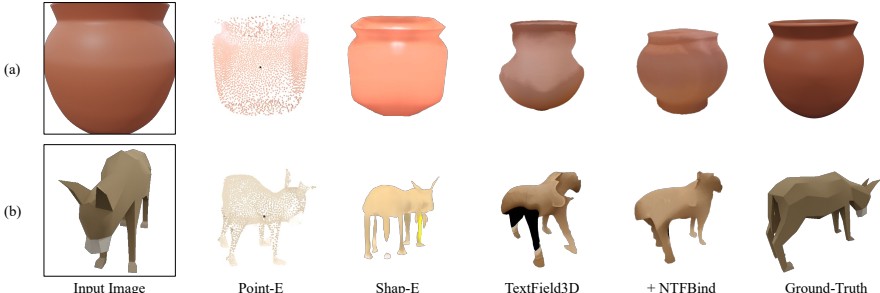

Figure 5: Image-to-3D generation. We use a clay pot in the bottom view and a donkey in the front view as image inputs. Our network can reproduce a close novel view to the ground-truth image.

aligning features in (a) by pre-trained models, which is similar to TAPS3D (Wei et al., 2023); (c) embedding text features with our NTFGen module; (d) further attaching our text-to-3D discriminator to (c), which is our method TextField3D. We train them on the ShapeNet dataset, with both BLIP-2 (Li et al., 2023) and MiniGPT-4 (Zhu et al., 2023) captioning.

We quantitatively compare the four solutions, as well as GET3D, in Table 2. FID (Heusel et al., 2017) and CLIP-score (Hessel et al., 2021) are used for the evaluation. The results demonstrate that our proposed modules effectively improve generation quality and text control consistency.

We also provide qualitative results in Figure 6. We use a simple text prompt "A green ball" to evaluate generation quality and a complicated text prompt "A round table with red legs and a blue

Table 2: Quantitative Analysis. FID (Heusel et al., 2017) and CLIP-score (Hessel et al., 2021) are adopted to evaluate the generation quality and text consistency of different solutions.

| Method | BLIP-2 (Li et al., 2023) | | | MiniGPT-4 (Zhu et al., 2023) | | |
|---|---|---|---|---|---|---|
| | FID↓ | CLIP-Score↑ | | FID↓ | CLIP-Score↑ | |
| | | ViT-B/32 | ViT-L/14 | | ViT-B/32 | ViT-L/14 |
| GET3D (Gao et al., 2022) | 41.66 | - | - | 41.66 | - | - |
| (a) | 31.67 | 28.33 | 22.79 | 34.67 | 28.59 | 23.44 |
| (b) | 29.02 | 29.80 | 24.39 | 29.91 | 30.03 | 24.05 |
| (c) | 29.73 | **30.36** | 25.43 | 28.69 | 30.68 | 25.17 |
| (d) | **26.94** | 30.35 | **25.57** | **25.46** | **30.89** | **25.77** |

"A green ball"          "A round table with red legs and a blue top surface"

(a)   (b)   (c)   (d)          (a)   (b)   (c)   (d)

Figure 6: Ablation studies. We use two text prompts to evaluate the generation quality and text control consistency of different solutions, demonstrating the effectiveness of our proposed modules.

top surface" to further evaluate text control consistency. In the former case, (d) presents a high-quality generation thanks to the multi-modal discrimination. In the latter case, (c) and (d) generate reasonable results according to the textual input, demonstrating the effectiveness of Noisy Text Field.

**NTFBind Module.** To evaluate view-invariant features $\widetilde{\mathbf{v}}_c^t = f_{view}(f_{vis}(I_c^t), c)$, we compare it with vanilla CLIP features $\mathbf{v}_c^t = f_{vis}(I_c^t)$. We propose two evaluation metrics: cosine similarity $s/\widetilde{s}$ and retrieval ratio $r/\widetilde{r}$. We randomly select front view $c_f$ and back view $c_b$ of 55 generations $I_{c_f}^t, I_{c_b}^t$ (one object for each category, with label $t$). For CLIP similarity, we calculate the mean similarities, i.e., $s = \sum_{i=1}^{55} cos(\mathbf{v}_{c_f}^{t_i}, \mathbf{v}_{c_b}^{t_i})/55$ and $\widetilde{s} = \sum_{i=1}^{55} cos(\widetilde{\mathbf{v}}_{c_f}^{t_i}, \widetilde{\mathbf{v}}_{c_b}^{t_i})/55$. For retrieval ratio, we regard the opposite view of the same object as the positive target and views of other objects as negative targets. $r$ and $\widetilde{r}$ are then computed as the average of retrieval ratios. As shown in Table 3, both the similarity and the ratio of view-invariant features are higher than CLIP features, which indicates that $L_{view}$ learns a view-invariant representation. It is more obvious in generated images, further demonstrating that view-invariant features enhance the generation consistency of novel views.

Table 3: Comparison of view-invariant features and vanilla CLIP features.

| Image Source | $s$ | $\widetilde{s}$ | $r$ | $\widetilde{r}$ |
|---|---|---|---|---|
| Ground-truth | 88.01% | 91.91% | 58.07% | 81.44% |
| Generated | 84.84% | 91.32% | 54.54% | 76.97% |

## 6 CONCLUSION

In this work, we present TextField3D to enhance the open-vocabulary capability of 3D generative models. Specifically, we introduce Noisy Text Fields (NTFs) to 3D latent code, enhancing the mapping of V-L pre-trained knowledge and 3D training data. An NTFGen module is proposed to generate noisy text latent code, and an NTFBind module is further proposed to bind view-invariant image latent code to NTFs. Furthermore, to supervise the generation quality and text consistency, we propose multi-modal discrimination that includes both text-3D and text-2.5D discriminators. TextField3D can efficiently generate various 3D contents with complicated text prompts, exhibiting a potential open-vocabulary generative capability.

**Broader Impacts.** 3D creations are in high demand for various applications. TextField3D can generate open-vocabulary 3D objects, greatly reducing the cost of producing 3D content. However, like other generative models, there may be a risk of generating malicious content.

**Limitations.** TextField3D still depends on the vocabulary of training data, which may not fully match the general capability of V-L supervised methods. For example, it fails in prompts like "A corgi is reading a book", as current 3D data mainly focuses on shape/texture-related concepts but lacks action-related prompts. Fortunately, the scale of 3D data is steadily increasing. More and more novel concepts can be included to expand TextField3D's vocabulary.

## ACKNOWLEDGEMENTS

This work was supported by National Key RD Program of China under Grant No. 2021ZD0112100, and the National Natural Science Foundation of China (NSFC) under Grant No. U19A2073.

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

## A   EXPERIMENTAL DETAILS

### A.1   EXPERIMENT IMPLEMENTATION

**NTFBind Training and Inference.**   To support image-to-3D generation, we additionally train an NTFBind module with the binding objective $\mathcal{L}_{bind}$. We load the parameters of the generator, the discriminator, and the NTFGen module on text-to-3D training. We freeze the NTFGen module during the training so that the NTFBind module can be aligned to the trained noisy text fields. The overall training objective $\mathcal{L}$ is similar to text-to-3D training:

$$\mathcal{L}_{image-to-3D} = \mathcal{L}(D_{img}, G) + \mathcal{L}(D_{mask}, G) + \lambda_{pc}\mathcal{L}(D_{pc}, G) + \lambda_{bind}\mathcal{L}_{bind}, \qquad (7)$$

where $\lambda_{bind}$ is set as $0.1$ in the experiment. Since the camera views are sampled uniformly in a fixed range, we can randomly generate a camera pose for an input image on inference.

**Ablation Studies.**   In the ablation studies of the main text, we compare four solutions for 3D conditional generation. We additionally illustrate the latent code encoder of solutions (a) and (b) in Figure 7. In (a), the latent code is initialized by a V-L pre-trained text encoder and supervised by a text-2.5D discriminator, the overall loss $\mathcal{L}_a = \mathcal{L}(D_{img}, G) + \mathcal{L}(D_{mask}, G)$. In (b), rendered images are further aligned with input text features, which can be regarded as $\mathcal{L}_{img}$ in our NTFGen module. Its overall loss can be revised as $\mathcal{L}_b = \mathcal{L}_a + \mathcal{L}_{img}$. In (c), a full version of our NTFGen module is adopted, in which the overall loss can be formulated as $\mathcal{L}_c = \mathcal{L}(D_{img}, G) + \mathcal{L}(D_{mask}, G) + \lambda_{bind}\mathcal{L}_{bind}$. Finally, (d) presents our TextField3D. Compared with (c), it is further supervised by a text-3D discriminator. Note that text features in (a)-(b) are fed to a mapping network, while we skip the mapping of text features in (c)-(d), as we mentioned in the main text.

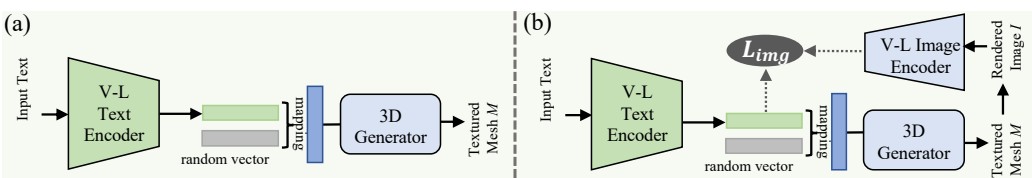

Figure 7: The latent code encoders of solutions (a) and (b) in the main text ablation studies.

## B   DATA PREPARATION

### B.1   IMAGE RENDERING

To generate 2D images from 3D meshes for training, we follow the rendering strategy in GET3D (Gao et al., 2022). We first scale each mesh object such that the longest edge of its bounding

box equals 0.8. We then render the RGB images and silhouettes from 8 random camera poses, which are sampled uniformly from an elevation angle of range $[0, \frac{1}{6}\pi]$ and a rotation angle of range $[0, 2\pi]$. For all camera poses, we use a fixed radius of 1.2 and the field of view (FOV) is $49.13°$. Images are rendered in Blender (Community, 2018) with fixed lighting.

## B.2 IMAGE CAPTIONING

To provide text-3D paired training data, we adopt two image captioning methods BLIP-2 (Li et al., 2023) and MiniGPT-4 (Zhu et al., 2023) to caption the rendered images in ShapeNet (Chang et al., 2015) and Objaverse (Deitke et al., 2022). Considering the data scale, we didn't use MiniGPT-4 to caption Objaverse. In this section, we will introduce the captioning strategies and compare the captioning effects through examples.

**BLIP-2.** BLIP-2 leverages a lightweight querying transformer to bridge the gap between pre-trained image encoders and language models. We use BLIP-2-OPT-2.7b as the captioning model. For each rendered image, BLIP-2 is employed to expand a sentence beginning with "*an object of*". We then extract {*text*} from the expanded sentence "*an object of* {*text*}" as the caption.

**MiniGPT-4.** MiniGPT-4 utilizes an advanced large language model (LLM) Vicuna (Chiang et al., 2023) and a pre-trained vision component of BLIP-2 to formulate multi-modal outputs. It aligns the encoded visual features with the Vicuna language model using just one projection layer and freezes all other vision and language components. Albeit a lightweight alignment, it can possess many similar capabilities to GPT-4, including generating intricate image descriptions and answering visual questions. Thus, given an input image and its category label {*c*}, we have a question template "*there is a 3D {c} in the picture, use one sentence to shortly describe its texture and shape*" for MiniGPT-4. Note that despite emphasizing "*use one sentence*", MiniGPT-4 is still likely to generate multiple sentences. To simplify captions, we use the first sentence of the answer as the final caption. Due to the time constraint, we only utilize MiniGPT-4 to make captions for ShapeNet in this paper.

**Comparison of Captioning Effects.** In Figure 8, we choose three representative images to compare the captioning effects of different methods. Since TPAS3D (Wei et al., 2023) also proposes a pseudo caption generation method for captioning rendered images in part of ShapeNet, we include it in the comparison. From the three cases, we can observe that captions generated by TAPS3D and BLIP-2 are relatively similar, primarily in terms of the combination of adjectives and nouns. In contrast, MiniGPT-4 provides detailed descriptions, including information in both texture and shape.

| | TAPS3D | BLIP-2 | MiniGPT-4 |
|---|---|---|---|
|  | "A red hatchback" | "A red sports car" | "The image is a red sports car with a sleek and shiny body, designed with a streamlined style" |
|  | "A brown chair" | "A wooden chair" | "The wooden chair has a rectangular shape with a curved back and a flat seat" |
|  | "A wooden desk" | "A wooden table" | "The image is a wooden table with a rectangular top and four legs" |

Figure 8: Comparison of image captioning effects.

### B.3 DATASET FILTERING

Although Objaverse (Deitke et al., 2022) is the largest public 3D dataset, its data quality varies. Therefore, we have filtered out some low-quality data. The filtering process has two steps: First, we filter out mesh files that contain more than one object, as our method is not suitable for multi-object/scene-level generation. Then, we filter out mesh objects according to roughness, because we find there are many objects with uneven surfaces, as well as objects that look like a piece of paper. We use Laplacian Regularizer (Hasselgren et al., 2021) to evaluate the roughness of an object, i.e., $\delta_i = v_i - \frac{1}{|N_i|} \sum_{j \in N_i} v_j$, where $N_i$ is the set of one-ring neighbours for vertex $v_i$. As the average laplacian regularizer metric $\overline{\delta}$ increases, the roughness level also increases. We select objects whose $\overline{\delta} \in [10^{-1}, 10^{-6}]$ as our final training data, which are not too flat or too rough for generation.

## C SUPPLEMENTARY RESULTS

### C.1 THREESTUDIO EVALUATION SET

We find a prompt list with 415 prompts in a public repo ThreeStudio and evaluate the CLIP R-Precision in this test set. Since we haven't surveyed any method that is evaluated in this set, we only compare our method with Shap-E. As shown in Table R3, although it is a harder test set, our method still significantly outperforms Shap-E.

Table 4: CLIP R-Precision on ThreeStudio evaluation prompts.

| Method | ViT-B/32 | ViT-L/14 | Latency | Data Scale |
|---|---|---|---|---|
| Shap-E (300M) | 13.98 | 19.51 | 1.0 V100-min | $> 1$M |
| TextField3D (ours, 1-shot) | 19.04 | 24.33 | $\sim 6.9$ V100-sec | $\sim 175$K |
| TextField3D (ours, 9-shot) | 37.83 | 42.41 | $\sim 1.0$ V100-min | $\sim 175$K |

### C.2 COMPARISON WITH MORE SOTA METHODS

The reason why not compare TAPS3D and SDFusion directly is that, they are fine-tuning methods based on single-category GET3D pre-training. For example, in the implementation of the original TAPS3D, a pre-trained category-wise GET3D checkpoint (e.g., car checkpoint, table checkpoint, and so on) is first loaded to the generator. The generator is then fine-tuned by the supervision of a fixed discriminator. Considering that GET3D hasn't been pre-trained in an open-vocabulary setting, it is hard to evaluate the original TAPS3D in our benchmark.

Nevertheless, we still provide a comparison based on their official code. We load the checkpoint of our pre-trained GET3D (Table 2) and further finetune the generator with TAPS3D. As shown in Table 5, its FID and CLIP-Score (ViT-B/32) results are uncompetitive to our method and even below the version we reproduced (named as (b) in the main text). We think the results are reasonable, as the performance of fine-tuned TAPS3D strongly depends on the pre-trained GET3D. We have demonstrated in Table 2 of the main text that GET3D can hardly generate high-quality 3D results in a multi-category dataset. As a result, TAPS3D is not an appropriate approach to open-vocabulary generation.

Furthermore, since TAPS3D hasn't released the training captions and finetuned checkpoints, we cannot even make a comparison on single-category generation with them. We notice that SDFusion has released its text-to-shape checkpoint, which is trained on a table-chair dataset. Therefore, we further evaluate the original SDFusion with recursive text prompts. As shown in Figure 10, SDFusion is a shape-only generator with low generation resolution. Besides, it is not sensitive to long sentences, failing to generate curved legs in the last prompt.

### C.3 ABLATION STUDY OF NOISY TEXT FIELD

To verify the noise range in our noisy text fields, we design four text prompts with gradually increased attributes for comparison with Shap-E. Jun & Nichol (2023) have mentioned that Shap-E struggles to bind multiple attributes to different objects. In Figure 9, it fails in the third case when

Table 5: Performance of the original TAPS3D. The results of GET3D and (b) are from Table 2 in the main text.

| Method | FID↓ | CLIP-score↑ |
|--------|------|-------------|
| GET3D | 41.66 | - |
| TAPS3D | 35.12 | 28.38 |
| (b) | 29.02 | 29.80 |
| ours | **26.94** | **30.35** |

two attributes are added. Shap-E generates extra chairs, despite the absence of such a requirement in the text input. And it is confused by the fourth case, generating blue legs and a mixed-color top. In contrast, TextField3D performs well in all cases and meanwhile presents a high-level generation diversity. Moreover, we calculated the variances of $\widetilde{\mathbf{t}}$ in the four cases, which are $\mathbf{0.2479}$, $\mathbf{0.2153}$, $\mathbf{0.1674}$, and $\mathbf{0.1355}$, respectively. The variances are consistent with our expectations of noise range, *i.e.*, a more detailed description has a smaller noise range.

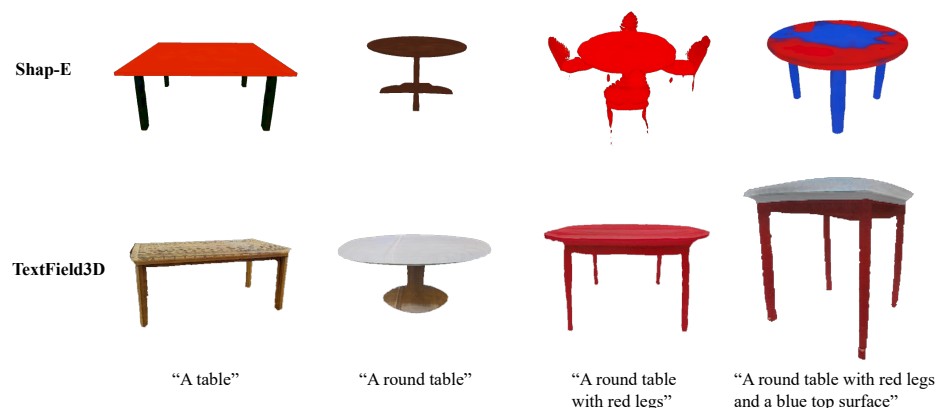

Figure 9: Examples of gradually increased attributes, comparing with Shap-E (Jun & Nichol, 2023).

To further evaluate the generation performance on gradually increased attributes, we include ShapeCraft (Fu et al., 2022) in the comparison. We use the recursive prompts in ShapCraft and rephrase them as longer sentences. As shown in Figure 10, TextField3D can achieve comparable performance with ShapeCraft, although it hasn't been specifically trained with recursive prompts. On the contrary, Shap-E only succeeds in generating the first case. It seems that Shap-E cannot understand "with **no** armrest", treating the phrase as "with armrest".

## C.4 ABLATION STUDY OF NTFGEN

Our NTFGen module consists of two objectives, *i.e.*, $\mathcal{L}_{img}$ and $\mathcal{L}_{txt}$. To verify the effectiveness of both objectives, we conduct two more experiments on each objective of ShapeNet (Chang et al., 2015) with captions from MiniGPT-4. We report the quantitative results in Table 6. Comparing two objectives, $\mathcal{L}_{img}$ has a better generation quality as its alignment of generations and text features. However, $\mathcal{L}_{txt}$ achieves a higher CLIP-score, which implies that $\mathcal{L}_{txt}$ can enhance the mapping of noisy text fields. Moreover, there's only one difference between "+$\mathcal{L}_{img}$" and solution (b) in the ablation studies of the main text, *i.e.*, the position of adding text features. "+$\mathcal{L}_{img}$" performs better than solution (b) (FID: 29.91, CLIP-score: 30.03) in terms of both FID and CLIP-score, demonstrating that bypassing the mapping network for text features is effective for generation quality and text control.

## C.5 ABLATION STUDY OF NTFBIND

Our NTFBind module binds image features in two dimensions: (1) to the noisy text fields; (2) to the multiple views. As illustrated in Table 7, two terms of binding are evaluated separately as "+binding

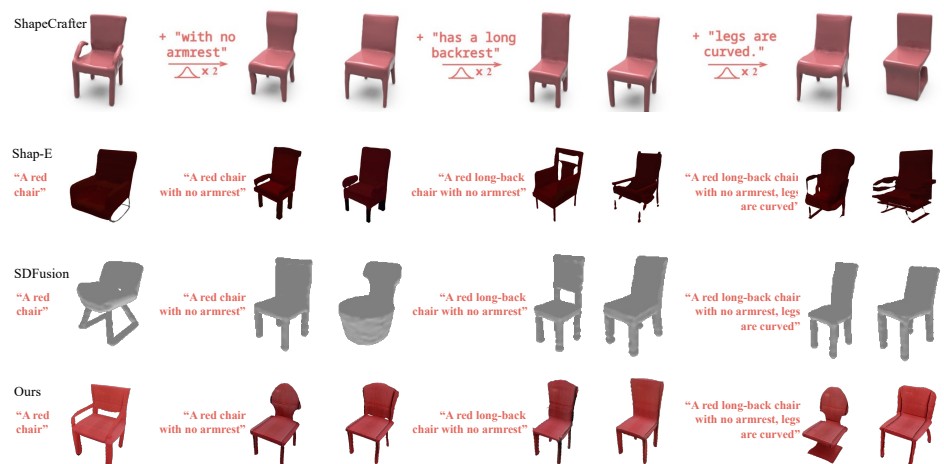

Figure 10: More examples of gradually increased attributes, comparing with ShapeCrafter (Fu et al., 2022), Shap-E (Jun & Nichol, 2023), and SDFusion Cheng et al. (2023).

Table 6: Quantitative results of NTFGen's ablation studies.

| Method | FID↓ | CLIP-score↑ |
|---|---|---|
| + $\mathcal{L}_{img}$ | 29.53 | 30.43 |
| + $\mathcal{L}_{txt}$ | 30.82 | 30.58 |
| + NTFGen | **28.69** | **30.68** |

NTFs" and "+binding views". It is obvious that binding NTFs can significantly improve the quality of image-to-3D generation (from 56.71 to 48.84, **-7.87** for FID). Although additionally adding a view-binding can further improve the FID to 48.07, we find that the training of binding-views alone does not lead to satisfactory results (we don't report its result as the model cannot even achieve convergence). We think it is caused by the disruption of alignment between the view-invariant latent code and the noisy text fields. Therefore, binding NTFs is necessary for the image-to-3D task.

Table 7: Quantitative results of NTFBind's ablation studies.

| Method | FID↓ |
|---|---|
| TextField3D | 56.71 |
| + binding NTFs | 48.84 |
| + binding views | - |
| + NTFBind | **48.07** |

## C.6 ABLATION STUDY OF MULTI-MODAL DISCRIMINATION

The intention of the discrimination design is to decouple the supervision of texture and shape generation. We mainly have two unsuccessful attempts: (1) we split the text condition and the camera condition into two independent discriminators, expecting that the former condition would focus on texture and the latter condition would focus on geometry. However, it fails to generate high-quality content (FID is 36.14). As a result, we combine two conditions together in our final discriminator. (2) we replace the mask discriminator with a depth discriminator, as depth provides more geometry information than the silhouette mask. However, all the generated objects fall into a sphere shape under depth discrimination. We think that is because depth discrimination is too sensitive to depth changes. Especially when using a mesh representation DMTet (Shen et al., 2021), the prominent vertices tend to be smoothed to the neighboring surfaces easily, formulating a sphere shape. Instead, we introduce a 3D discriminator based on point cloud, which can flexibly represent the mesh surface.

## C.7 MORE VISUALIZATION RESULTS

To better present our open-vocabulary generation capability, we visualize more generated objects in Figure 11 and 12. We also provide multiple views of these objects in Figure 13. The elevation angles for rendering are $-15°$, $15°$, and $30°$ from left to right, respectively. In Figure 14, we generate 9 examples for each prompt, showcasing TextField3D's generation diversity. Note, a beer can in our training dataset is given in the left of Figure 14 to demonstrate that our results don't overfit to training data. Furthermore, we provide more visualization results of complicated input prompts and conditioned images in Figure 15 and Figure 16, respectively.

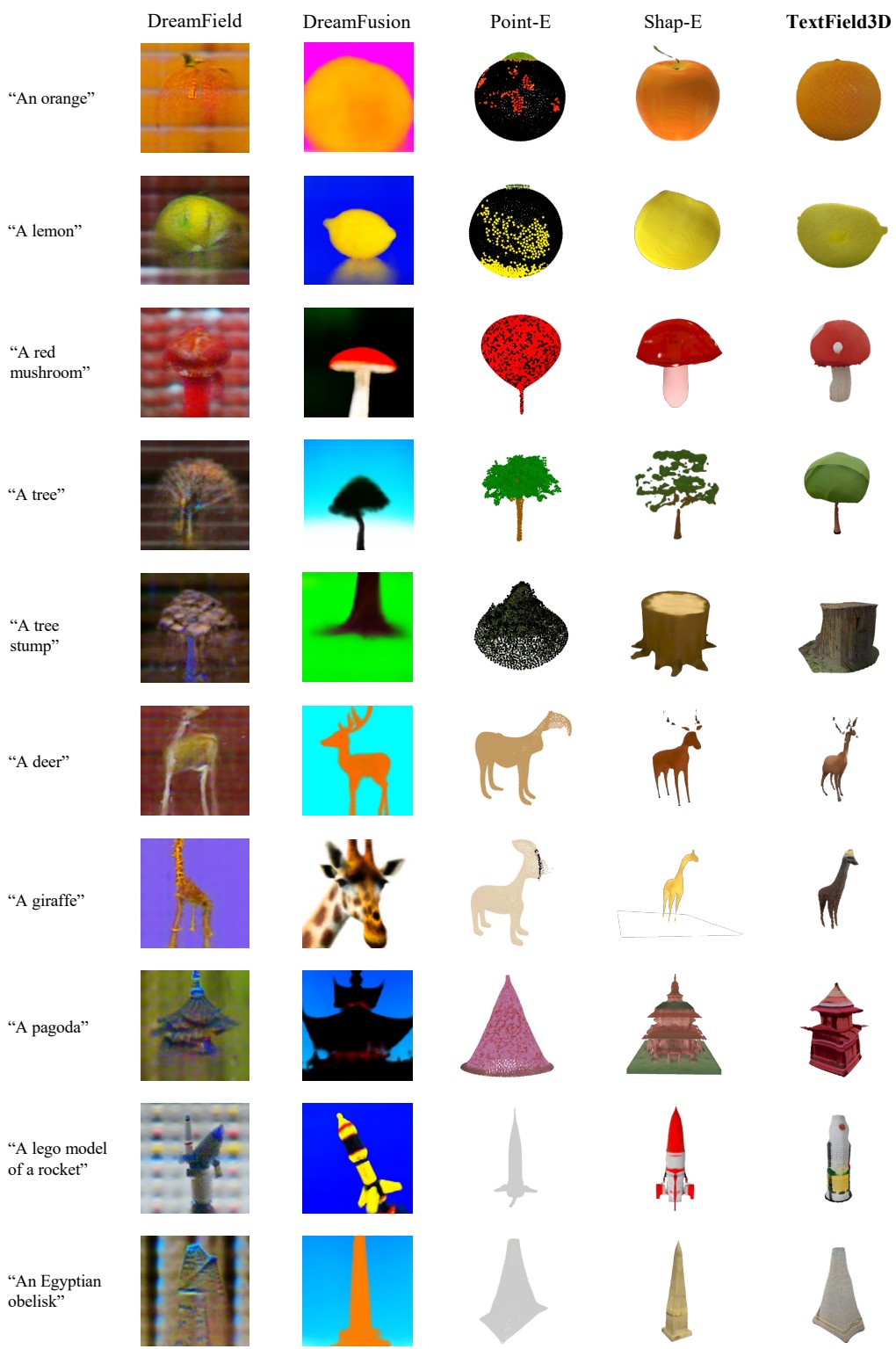

Figure 11: More visualization results, comparing with previous state-of-the-art methods.

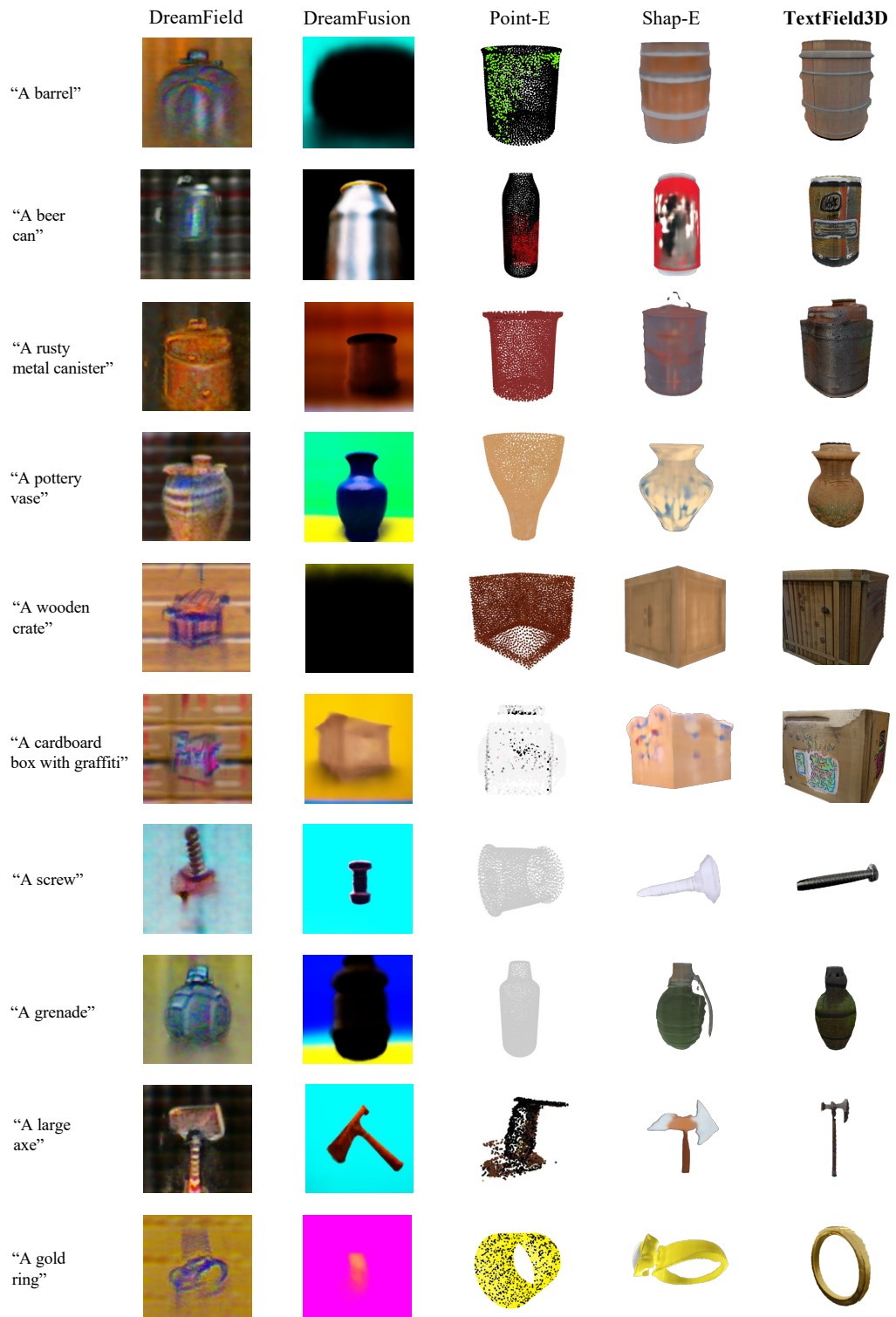

Figure 12: More visualization results, comparing with previous state-of-the-art methods.

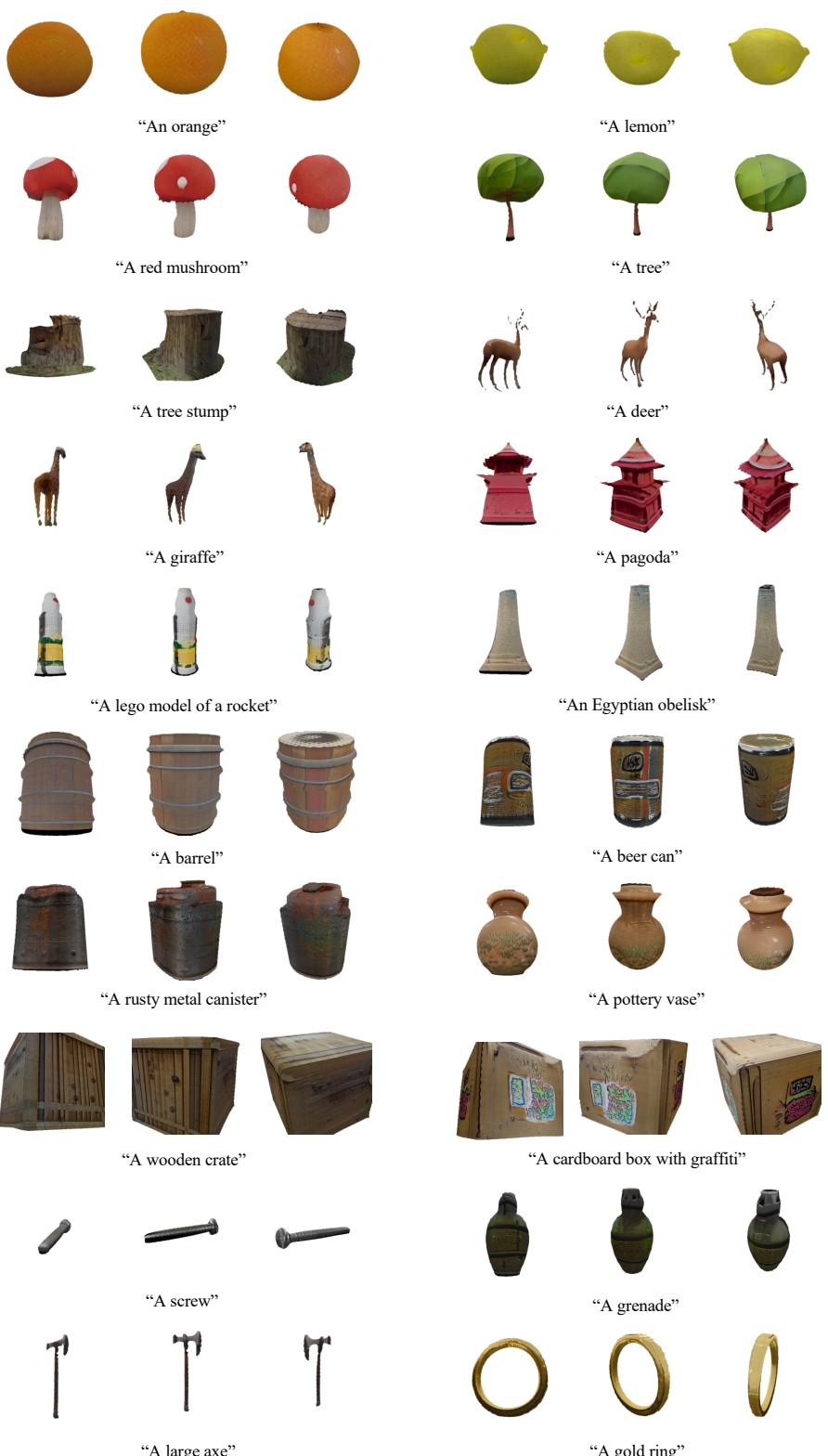

Figure 13: Multi-view visualization of generated objects in Figure 11 and 12. The elevation angles for rendering are $-15°$, $15°$, and $30°$ from left to right, respectively.

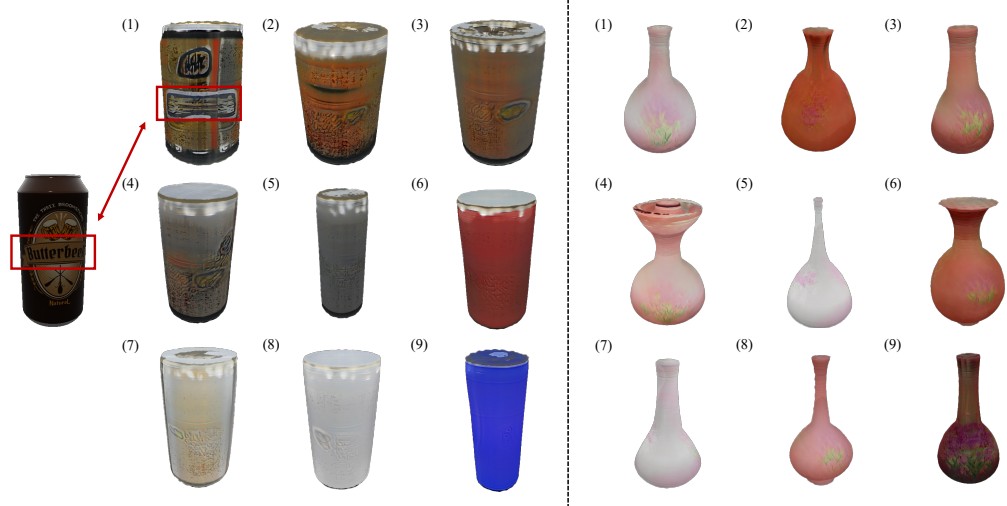

Figure 14: 9-shot examples. We provide 9 different bear cans, presenting the diversity of our generation in a given text. The left beer can is the most similar one we detect in our training dataset, demonstrating that our method hasn't overfitted to training data.

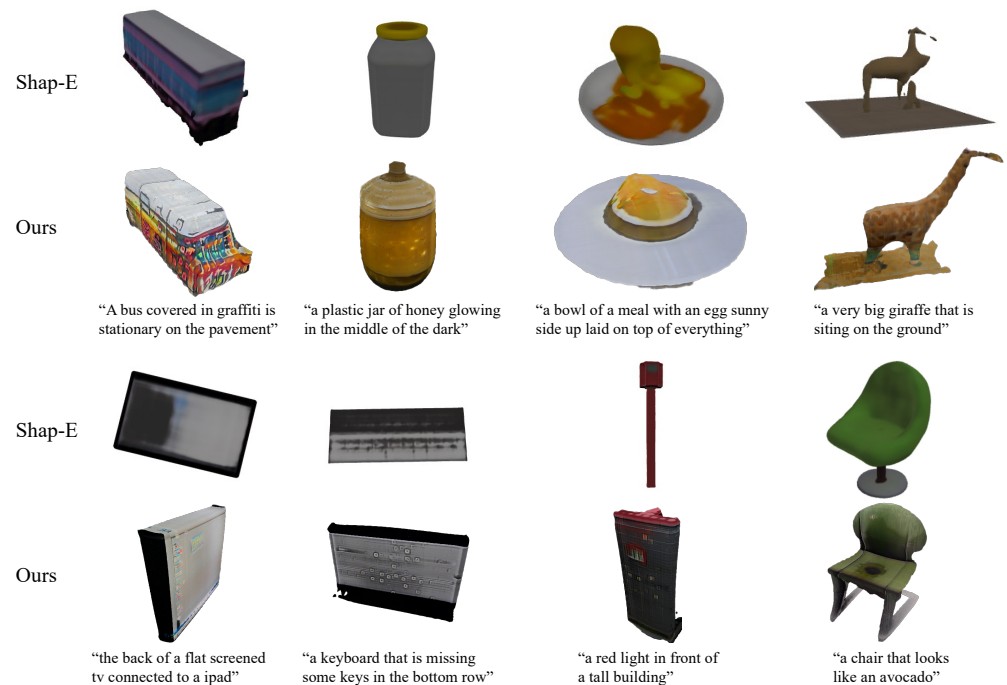

Figure 15: Complicated prompts. To further present the open-vocabulary capability, we provide the visualization results of our generation in complicated prompts. Shap-E is included in the comparison.

Input

Shap-E

Ours

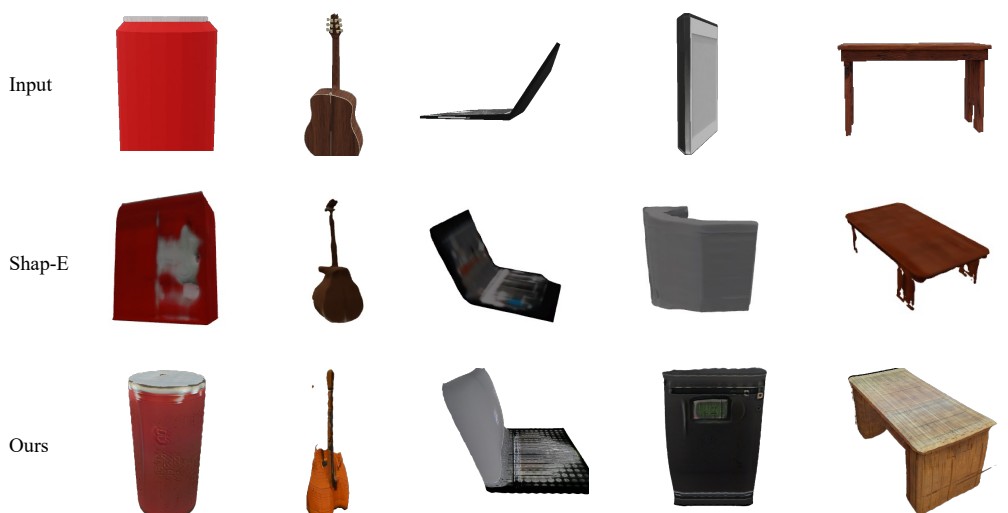

Figure 16: Image-to-3D generation. We provide more visualization results of text-to-3D generation. Shap-E is included in the comparison.

