# OpenReview forum: "TextField3D: Towards Enhancing Open-Vocabulary 3D Generation with Noisy Text Fields"
_ICLR.cc/2024/Conference — ICLR 2024 poster_

### Official Review · Reviewer_9TYY · 2023-10-27

**Soundness:** 3 good
**Presentation:** 3 good
**Contribution:** 3 good
**Rating:** 6
**Confidence:** 4

**Summary:**

This paper aims to achieve open-vocabulary 3D generation by attempting to tackle the challenges of limited 3D data and therefore sparse text annotations, which would limit the model's ability to generalize to open-vocabulary queries. One way to approach the challenge is to add noises to text features to avoid models from overfitting. But it is not trivial to know how much noise is appropriate. Therefore, the authors propose to learn a noisy text fields, which learns the standard deviation of the Gaussian noise that can be added to each text feature. Other than the text-to-image task, this paper also proposes to learn a view-invariant image representation to facilitate better image-to-3D generation. As for the generator part, the authors use a 3D-aware GAN framework, with a GET3D-like generator and a text-3D and text-2.5D discriminator. Qualitative and quantitative experiments are reported to showcase the performance of the proposed method. For the quantitative part, TextFireld3D surpasses baseline methods, including Point-E, Shap-E and GET3D. Visual results also show reasonable generation quality both from text and image conditions.

**Strengths:**

1. Overall, I like that the authors choose to use 3D-aware GAN instead of diffusion models to tackle the text-to-3D generation problem. GAN has clear advantages over diffusion model in terms of generation speed and smooth interpolation. But for the open-vocabulary text-guided generation task, GAN falls behind diffusion models. Therefore, it is exciting to see a GAN model surpasses diffusion baselines, e.g., Shap-E and Point-E. This paper can service as a strong baseline for the community of text-to-3D generation.
2. The idea of learning the noisy text field is interesting. Experiments also suggest the effectiveness of this method.

**Weaknesses:**

My main concerns are around the experiments of this paper.

1. I would expect to see more baseline methods being compared with. For example, in the ablation study, authors mentioned methods like SDFusion and TAPS3D.
2. I would suggest also testing with the DreamFusion testing list, which contains over 400 prompts.
3. For the image-to-3D generation, the evaluation is limited to showcasing two visual results, which is far from enough. I would suggest adding more visual results, especially showing multiple viewing angles of the generated object. Some quantitative evaluation is also expected.
4. For the visual results, I would like to see more prompts from the DreamField or DreamFusion test sets, which include more challenging and complex examples like concept-mixing. It would help us better evaluate the open-vocabulary ability of the proposed method. Right now, the showcasing prompts only contain one simple concept and can be easily found in Objaverse training set. I understand that concept mixing is super hard given such limited training data. But at least, it is nice to see some failure cases and analysis on the failure modes.

If authors can provide more evaluations during the rebuttal. I would consider raising the score.

**Questions:**

See weaknesses.

---

> ### Author Response · Authors · 2023-11-20
>
> We thank the reviewer for the constructive comments and the opportunity to clarify our approach.
>
> ### Q1. More Baselines
> The reason why not consider more baselines is that few available 3D-supervised methods are designed towards open vocabulary 3D generation, except Point-E and Shap-E.
> We explained in Q4 of Reviewer oUD4 that available TAPS3D requires a pre-trained GET3D checkpoint, which is supervised in a single-category manner. Similarly, SDFusion has to pre-train a VQVAE, while limited categories are included in its pre-training dataset.
>
> Nonetheless, we still pre-trained a GET3D in our ShapeNet55 dataset (already done in Table 2 of the main text) and attempted to fine-tune it with TAPS3D. As shown in Table R2 of Reviewer oUD4, the generation quality is restricted by the pre-trained GET3D, falling below our method. Considering GET3D can hardly generate high-quality 3D results in a multi-category manner, the fine-tuning strategy in TAPS3D is not an appropriate approach to open-vocabulary generation. SDFusion may meet a similar situation to TAPS3D.
>
> One can only compare our method with the original SDFusion/TAPS3D in a single-category generation task. SDFusion has released its text-to-shape checkpoint, which is trained on a table-chair dataset. Therefore, we update Figure 10 in Appendix, further evaluating the original SDFusion with recursive text prompts. As shown in Figure 10, SDFusion is a shape-only generator with low generation resolution. Besides, it is not sensitive to long sentences, failing to generate curved legs in the last prompt.
>
>
> ### Q2. DreamFusion Test List
> We reviewed the original paper of DreamFusion and didn't find the testing list with over 400 prompts. The paper mentions that "*we use the 153 prompts from the object-centric COCO validation subset of Dream Fields*", which is exactly the set we used in our paper. However, we find a prompt list in a public repo [threestudio](https://github.com/threestudio-project/threestudio/blob/main/load/prompt_library.json), which includes 415 prompts. We evaluate the CLIP R-Precision in this test set. Since we haven't surveyed any method that is evaluated in this set, we only compare our method with Shap-E. As shown in Table R3, although it is a harder test set, our method still significantly outperforms Shap-E.
>
> Table R3. CLIP R-Precision results on a test list with over 400 prompts.
> |CLIP R-Precision|ViT-B/32|ViT-L/14|
> |:----:|:----:|:----:|
> |Shap-E (300M)|13.98|19.51|
> |Ours, 1-shot|19.04|24.33|
> |Ours, 9-shot|37.83|42.41|
>
>
> ### Q3. More Image-to-3D Results
> We provide more visualization examples in Figure 16 of the updated Appendix. Compared with Shap-E, our generation results present more detailed textures and more reasonable geometries. As for the quantitative evaluation, we have provided the ablation study of NTFBind in Table 5 of Appendix, which verifies our improvement in FID. We further evaluate Shap-E in this test set, and its FID is 51.32. The quantitative result is consistent with the visualization comparison. We will update the result in our revision.
>
>
> ### Q4. More Complex Prompts
> We select some complicated prompts from DreamField/DreamFusion evaluation sets and compare our results with Shap-E in Figure 15 (in the updated Appendix). Our results basically contain more detailed textures and more reasonable geometries when compared with Shap-E, exhibiting stronger open-vocabulary capability. As the reviewer mentioned in Weakness 4, concept mixing is super hard given such limited training data.
> Our method can understand most of the descriptions in the given prompts. While we acknowledge that the quality of our generation results is still not satisfactory enough. The generation quality compromises as the descriptions are increased.
> For example, the *giraffe* in Figure 15 is not as complete as the one generated in Figure 11. And our method is limited in imagination. As for the *chair that looks like an avocado*, our generated result simply combines the shape of a chair with the color of an avocado. Nonetheless, it is crucial to acknowledge the substantial contributions our work has made in advancing open-vocabulary text-to-3D with limited data.
> We will add an additional discussion in the limitation part of our revised manuscript.

---

> ### Author Response · Authors · 2023-11-22
> **Looking forward to further discussions**
>
> Dear reviewer,
>
> Thank you again for your insightful comments on our paper, and we genuinely hope that our response could address your concerns. As the discussion is about to end, we are sincerely looking forward to your feedback. Please feel free to contact us if you have any further inquiries.

---

> > ### Comment · Reviewer_9TYY · 2023-11-22
> >
> > Thank you for the detailed rebuttal and additional experimental results. Most of my concerns are addressed. I would raise my score. Looking forward to see these experiments and discussion added to the final revision.

---

> > > ### Author Response · Authors · 2023-11-23
> > >
> > > Thank you for the positive feedback on our work! Your constructive suggestions are very helpful for us to improve the paper. We will include new experiments and discussions in our final revision.

---

### Official Review · Reviewer_oUD4 · 2023-10-30

**Soundness:** 2 fair
**Presentation:** 4 excellent
**Contribution:** 2 fair
**Rating:** 6
**Confidence:** 5

**Summary:**

This paper works on text-to-3d shape generation. The paper observes that the current text-to-shape generation methods are usually V-L Optimized or 3D supervised. These methods either suffer from the problem of long-optimization or restricted open-vocabulary capability. To resolve these problems, the paper proposes TextField3D, which generates 3D shapes in real-time, taking both open-vocabulary text prompts and image as input. Specifically, the method adopts a NFGen model, that maps a single text prompt into a noisy text field. This noisy text field enables more open-vocabulary text prompts than single category name or template-generated text prompts. It also proposes an NTFBind model, which maps images from any view into a view-invariant image feature in the noisy text field. The noisy latent feature from the noisy text field is then fed into a conditional generative model. The method also adopts a 2.5D and a 3D supervision that ensures the generated shape has high-quality texture and geometry. Experiment shows the proposed method generates shapes of higher quality than some V-L optimized and 3D supervised methods. It also shows the effectiveness of the NTFGen and NTFBind modules in design choice.

**Strengths:**

+ The paper is clearly written. The motivation is clear to me - design a real-time supervised method whose performance is competitive with VL-optimized methods.
+ The NTFGen model shows that it is able, to some extent, to increase the expressiveness of a single text prompt.
+ The NTFBind model shows that it is better at aligning image features from any views to text features.
+ The generated shapes visually seem to have better quality and texture than the compared methods thanks to the supervision from 2.5D and 3D.

**Weaknesses:**

This paper fails to convince me of the effectiveness of its major component in the following aspects:
+ Shape Diversity is limited by the training dataset. Quoting from the original paper - "With limited 3D data, can we train a real-time generator that is equivalent to V-L optimized methods", I think this is impossible considering the method is training with a relatively small scale 3D shape dataset compared with VLMs. VL-optimized methods clearly can generate synthesized imagined shapes, like a chair with the shape of an avocado, but given the qualitative examples that the authors provide, this method seems not able to generate imagined shapes. Even though I don't think this is a major drawback of the proposed method, I think this claim is faulty.
+ Open-vocabulary capability - The open-vocabulary capability is the major claim of this paper. However, in the qualitative experiment section, the paper only provides very simple prompts, like category names, adjective nouns, or a phrase with two nouns. I think these simple phrases are not complicated enough to prove the open-vocabulary capability of the method, especially considering the method is training with complicated enough captions generated by BLIP-2 or MiniGPT4. I hope the authors can provide more results from complicated text prompts as in the captions generated.
+ View-invariant experiments. The paper claims that the NTFBind model produces a view-invariant feature, but the experiment provided is not strong enough to prove the point.  The experiment uses image features across views and image features across ShapeNet categories to prove that image features across views are more reassembled than features across categories. I think this setting is not strong and persuasive enough. A better experiment setup would be comparing features across views with features across instances in the same category. See CLIP-NeRF[1] Figure 2 for more details.
+ Comparison. Though the paper compared with VL optimized methods and 3D supervised methods, it didn't directly compare with the TAPS3D, which has the same training setting as the proposed method. Both of them methods use an image captioning model to augment text prompts and work on a 3D-generated model. Though the paper provides an ablation study that replaces the major component to the TAPS3D component, I wonder if it will outperform the TAPS3D original method.
[1] CLIP-NeRF: Text-and-Image Driven Manipulation of Neural Radiance Fields

**Questions:**

+ Some minor questions:
1. For noisy text latent code, the paper uses a learned noise, which is referred to as "dynamic noise". I wonder how the method performs with a non-dynamic noise, which could set \sigma to a static number.
2. The noting of L_{img} and L_{txt} looks like two different types of loss, but they are actually the same type of loss. Changing the namings to make them more consistent would be better for reading.
3. Textured Mesh Generator. Are they training from scratch, or training from the pre-trained GET3D?
4. Sillhoutte loss. The silhouette loss in equations (4) and (5) is not introduced clearly. For a first-time reader, it might be a little confusing.

---

> ### Author Response · Authors · 2023-11-20
>
> We thank the reviewer for the constructive comments.
> ### Q1. Diversity Claim
> We acknowledge the reviewer's point regarding the challenge of achieving shape diversity with a smaller scale 3D shape dataset in comparison to 2D diffusion models. The term "equivalent" was perhaps optimistic, and we have amended our language to more accurately reflect our findings.In Table 1 of the main text, our CLIP R-Precision results significantly outperform previous 3D supervised methods and even get close to V-L pre-trained methods.
> The results indicate that our method exhibits its potential towards open-vocabulary generation. Accordingly, we revise the expression here as: *With limited data, can we train a real-time 3D generator with the potential towards open-vocabulary content creation?*
> ### Q2. Open-Vocabulary Capability
> We select some complicated prompts from DreamField/DreamFusion evaluation sets and compare our results with Shap-E in Figure 15 (in the updated Appendix). Our results basically contain more detailed textures and more reasonable geometries when compared with Shap-E, exhibiting stronger open-vocabulary capability. As Reviewer 9TYY mentioned in Weakness 4, concept mixing is super hard given such limited training data. Our method can understand most of the descriptions in the given prompts. While we acknowledge that the quality of our generation results is still not satisfactory enough. The generation quality compromises as the descriptions are increased. For example, the *giraffe* in Figure 15 is not as complete as the one generated in Figure 11. And our method is limited in imagination. As for the *chair that looks like an avocado*, our generated result simply combines the shape of a chair with the color of an avocado. Nonetheless, it is crucial to acknowledge the substantial contributions our work has made in advancing open-vocabulary text-to-3D with limited data. We will add an additional discussion in the limitation part of our revised manuscript.
>
>
> ### Q3. View-Invariant Experiments
> It is reasonable to compare features in the same category. We randomly select 50 instances in each category and calculate the average retrieval precision of all 55 categories in Table R1. Our view-invariant features still present an obvious improvement against vanilla CLIP features. Note that comparing different instances in the same categories is a more challenging task, as different instances in the same view sometimes share more similar content than the same instance in different views. We will update the results in our ablation study. Thanks for the suggestion.
>
> Table R1. Comparison of view-invariant and vanilla CLIP features across instances in the same category. $r$ is the retrieval precision with CLIP features and $\widetilde{r}$ is the retrieval precision with our view-invariant features.
> |Image Source|$r$|$\widetilde{r}$|
> |:----:|:----:|:----:|
> |Ground-Truth|45.60%|59.04%|
> |Generated|44.84%|57.34%|
> ### Q4. Comparison with TAPS3D
> The reason why not compare TAPS3D is that it is a fine-tuning method based on single-category GET3D pre-training. In the implementation of the original TAPS3D, a pre-trained category-wise GET3D checkpoint (e.g., car checkpoint, table checkpoint, and so on) is first loaded to the generator. The generator is then fine-tuned by the supervision of a fixed discriminator. Considering that GET3D hasn't been pre-trained in an open-vocabulary setting, it is hard to evaluate the original TAPS3D in our benchmark.
>
> Nevertheless, we still provide a comparison based on their official code. We load the checkpoint of our pre-trained GET3D (Table 2 in the main text) and further finetune the generator with TAPS3D. As shown in Table R2, its FID and CLIP-Score (ViT-B/32) results are uncompetitive to our method and even below the version we reproduced (named as (b) in the main text). We think the results are reasonable, as the performance of fine-tuned TAPS3D strongly depends on the pre-trained GET3D. We have demonstrated in Table 2 of the main text that GET3D can hardly generate high-quality 3D results in a multi-category dataset. As a result, TAPS3D is not an appropriate approach to open-vocabulary generation.
>
> Furthermore, since TAPS3D hasn't released the training captions and finetuned checkpoints, we cannot even make a comparison on single-category generation with them. We notice that SDFusion has released its text-to-shape checkpoint, which is trained on a table-chair dataset. Therefore, we update Figure 10 in Appendix, further evaluating the original SDFusion with recursive text prompts. As shown in Figure 10, SDFusion is a shape-only generator with low generation resolution. Besides, it is not sensitive to long sentences, failing to generate curved legs in the last prompt.
>
> Table R2. Performance of the original TAPS3D. The results of GET3D and (b) are from Table 2 in the main text.
> |Method|FID|CLIP-Score|
> |:----:|:----:|:----:|
> |GET3D|41.66|-|
> |TAPS3D|35.12|28.38|
> |(b)|29.02|29.80|
> |Ours|26.94|30.35|

---

> ### Author Response · Authors · 2023-11-20
> **Replies to the extra questions**
>
> ### Q5. Dynamic Noise
> We have tried this non-dynamic noise before. We set $\sigma$ to a static number 0.016, which is exactly the setting in [A]. We trained the non-dynamic noise in ShapeNet55 dataset with BLIP-2 captions, which is evaluated with 27.64 in FID and 30.06 in CLIP-Score (ViT-B/32). The results are not competitive to our dynamic noise (26.94 in FID and 30.35 in CLIP-Score). We will add the results to Appendix.
>
>
> ### Q6. Naming of $L_{img}$ and $L_{txt}$
> Thanks for the suggestion. We rename $L_{img}$ and $L_{txt}$ as $\mathcal{L}\_{nce}^{I}$ and $\mathcal{L}\_{nce}^{T}$, respectively. The revised names can be more consistent in meaning.
>
>
> ### Q7. Textured Mesh Generator
> We train the generator from scratch. We aim to train a unified generator for multiple categories, while GET3D aims to train category-wise generators. Loading its pre-trained model is not beneficial for our training.
>
>
> ### Q8. Silhouette Loss
> Thanks for the suggestion. We will provide an additional explanation in our revision that the silhouette loss is followed from GET3D.
>
> [A] Text-Only Training for Image Captioning using Noise-Injected CLIP.

---

> ### Author Response · Authors · 2023-11-22
> **Looking forward to further discussions**
>
> Dear reviewer,
>
> Thank you again for your insightful comments on our paper, and we genuinely hope that our response could address your concerns. As the discussion is about to end, we are sincerely looking forward to your feedback. Please feel free to contact us if you have any further inquiries.

---

> ### Comment · Reviewer_oUD4 · 2023-11-22
> **Comments regarding rebuttal**
>
> I appreciate the author's effort to include more results. The new multi-view experiment, explanation with TAPS3D, and the ablation on dynamic noise prove the technical solidity of the paper.
>
> For the open-vocabulary results, from visual perception results, I think the generated shape indeed has higher texture quality, but the shape does not semantically match with the textual input as Shape E. I acknowledge this is a difficult problem using any CLIP-like loss as supervision. I suggest the author include it in the limitation.
>
> Overall, I think the paper is technically solid. The NTFGen and NTFBind are effectively proved by the added experiments. I recommend acceptance of this paper.

---

> > ### Author Response · Authors · 2023-11-23
> >
> > Thank you for the positive feedback on our work! Your constructive suggestions are very helpful for us to improve the paper. We agree with your point that the generated shape may not be well matched with the open-vocabulary textual prompts. One possible solution is to enhance our point supervision in terms of text consistency, and we may consider that in future work.  We will discuss the limitation of open-vocabulary shape results in our final revision.

---

### Official Review · Reviewer_Fjdj · 2023-11-01

**Soundness:** 3 good
**Presentation:** 3 good
**Contribution:** 3 good
**Rating:** 8
**Confidence:** 3

**Summary:**

In this paper, the authors present TextField3D, which is a conditional 3D generative model that enhances text-based 3D data generation by injecting dynamic noise into the latent space of text prompts, creating Noisy Text Fields (NTFs). This technique allows for the mapping of limited 3D data to a broader range of textual latent space, enhanced by the NTFs. The authors propose two modules to facilitate this process: NTFGen, which models the general text latent code in noisy fields, and NTFBind, which aligns view-invariant image latent code to the noisy fields, aiding image-conditional 3D generation. The model is guided through the conditional generation process in terms of both geometry and texture by a multi-modal discrimination setup, consisting of a text-3D discriminator and a text-2.5D discriminator. TextField3D is highlighted for its large vocabulary, text consistency, and low latency, positioning it as an advancement over previous methods in the field.

**Strengths:**

- The paper successfully expands the GAN-based GET3D framework to handle extensive vocabulary datasets, achieving results on par with or surpassing those of diffusion-based models like point-e and ShapE. This marks a significant step forward for large-vocabulary feed-forward generative models.
- Since text and 3D are not one-to-one mapping, , the introduction of Noisy Text Fields and their corresponding modules seem to be reasonable to me.
- The results, both qualitative and quantitative, look diverse and of a relatively good quality compared to other feed-forward models.
- The ablation studies are clear and comprehensive, providing detailed insights into the impacts of various modules, discriminators, and choices in noise range.

**Weaknesses:**

- My main concern is the potential overfitting problem. In Figure 12 and 13, certain prompts (e.g., "A beer can", "A wooden crate", and "A cardboard box with graffiti") generate unusually detailed outputs, showing a much higher level of details than others. Based on my experience,  the training dataset likely contains very similar examples. I am interested in understanding how the authors have addressed and evaluated the risk of overfitting associated with their method.

**Questions:**

1. I'm wondering how is the FID score calculated under the text-conditioned setting?
2. Can you provide some more examples of the 9-shot experiments?

---

> ### Author Response · Authors · 2023-11-20
>
> Thank you very much for the encouraging and constructive comments.
>
> ### Q1. Potential Overfitting
> Overfitting in generative models is a complicated issue that is worth investigating. In [A], overfitting is defined as "*containing an object (either in the foreground or background) that appears identically in a training image, neglecting minor variations in appearance that could result from data augmentation*". They detect and analyze content replication in the diffusion synthesis results, concluding that most of the generated images do not contain significant copied content, but a non-trivial amount of copying does occur.
>
> Following [A], we detect the most similar object to our generated *beer can* among the Objaverse dataset, which is shown on the left of Figure 14 in the updated Appendix.
> Our generated *beer can* only shares a similar rectangular text box with it.
> Similar to the conclusion in [A], our text box is a regional copy, rather than a significant content replication.
> The same phenomenon can also be observed in the graffiti of the bus in Figure 15, which is similar to the graffiti on the cardboard you mentioned.
> Therefore, those unusually detailed outputs may be a possible regional copy but are not exactly the overfitting problem.
>
> To some extent, our NTFGen module is beneficial for alleviating overfitting, as the injected noise enlarges the text input to a range of objects.
> As shown in Figure 14 (in the updated Appendix), we provide 9-shot examples of our generated *beer cans* and *vases*. Our results present various textures. Few of them share a similar color or texture.
> We will update the discussion of overfitting in the Appendix accordingly.
>
> ### Q2. FID Score
> For 3D objects in the evaluation set, we have corresponding captions and ground-truth rendered images $\hat{I}$. To evaluate the FID metric in the text-to-3D task, we generate 3D content according to captions and then render generated images $I$. We compute the FID between the ground-truth set {$\hat{I}$} and the generated set {$I$}.
>
> ### Q3. 9-Shot Experiments
> Our 9-shot experiment is to generate an object nine times and choose the best generation based on CLIP similarity. We include this experimental setting because our generation process is significantly faster than other methods. As an example of visualization results, we update nine beer cans and vases in Figure 14. Our generation results vary greatly in geometry and texture, presenting our diversity in a given text prompt. In the 9-shot experiment, we chose the most similar rendered image (the first one, based on CLIP similarity) for calculating R-Precision.
>
> [A] Diffusion Art or Digital Forgery? Investigating Data Replication in Diffusion Models.

---

> ### Author Response · Authors · 2023-11-22
> **Looking forward to further discussions**
>
> Dear reviewer,
>
> Thank you again for your insightful comments on our paper, and we genuinely hope that our response could address your concerns. As the discussion is about to end, we are sincerely looking forward to your feedback. Please feel free to contact us if you have any further inquiries.

---

### Meta-Review · Area_Chair_AzDu · 2023-12-12

**Metareview:**

The paper has received unanimous acceptance recommendations with scores 6/6/8, with all reviewers agreeing that it enhances GAN-based fast 3D generation approaches by enabling open-vocabulary generation. The authors have successfully addressed most concerns raised in the original reviews through the rebuttal process. While reviewer oUD4 pointed out that the open-vocabulary capability of the proposed method is still inferior to V-L methods, the authors have incorporated this limitation into the manuscript.
After a thorough evaluation, the Area Chair (AC) does not find sufficient evidence to overturn the reviewers' positive assessments. Consequently, the AC recommends accepting the paper.

**Justification For Why Not Higher Score:**

While the paper advances the GAN-based 3D generative model, the quality of generated 3D models is still much inferior to the concurrent V-L methods. The method is still limited by the available 3D training data, which can not be scaled easily. Therefore, AC decides not to give a higher score for this work.

**Justification For Why Not Lower Score:**

All reviewers recommend accepting the paper, and this paper does make an advancement on GAN-based 3D generative modeling approaches. Therefore, AC recommends accepting the paper.

---

### Decision · Program_Chairs · 2024-01-16

Accept (poster)